aquaculture; fisheries; wellbeing; fairness; equity

**Corresponding author:**
Karen A. Alexander;
Email: karen.alexander@hw.ac.uk

# Social sustainability in seafood systems: a rapid review

Karen A. Alexander[1]  and Ingrid Kelling[2]

[1]International Centre for Island Technology, Institute of Life and Earth Sciences, School of Energy, Geoscience, Infrastructure and Society, Heriot-Watt University, Stromness, UK and [2]The Lyell Centre, Institute of Life and Earth Sciences, School of Energy, Geoscience, Infrastructure and Society, Heriot-Watt University, Edinburgh, UK

## Abstract

Sustainability and sustainable development are the buzzwords of our era. Nowhere is this clearer than in primary production/extraction industries, such as aquaculture and fisheries. Yet in the seafood sector (as with many others), the term continues to be used most commonly in relation to the environmental dimension; much less is known about social and economic sustainability. In this review, we explore what is known about social sustainability in the seafood sector. We identify seven key thematic areas: livelihoods and human development; human rights; social, psychological, and cultural needs; equitable access to resource and benefit sharing; a voice in public issues; flow-on benefits for local and regional economies and improved infrastructure and access. We reveal that while there has been a clear focus on developing social sustainability indicators, this has largely missed more relational and subjective aspects of social sustainability. We also show that some thematic areas of social sustainability also remain underdeveloped. Overall, we argue that it is imperative that we address the knowledge gaps and incorporate what we already know about social sustainability into existing industry and governance processes. If we do not, not only risk not achieving the Sustainable Development Goals, but we also risk moving closer towards environmental and societal collapse.

## Impact statement

Despite a rise in emphasis on the social mandate of sustainable development discourse, there is much uncertainty regarding the many meanings and applications of the term 'social sustainability'. This has meant that, like other industries, the seafood sector has been criticised for neglecting social issues. In this review, we provide a broad overview of the current state of knowledge relating to social sustainability in the seafood sector (comprising fisheries and aquaculture). We also identify where research gaps remain. We also propose means by which social sustainability can be incorporated into existing industry and governance processes. We anticipate that this review will be of benefit in two ways to inform: i) those working in social sustainability in the seafood sector and associated organisations regarding potential areas on which to focus their efforts and ii) scholars regarding directions for future research.

## Introduction

Sustainability and sustainable development are the buzzwords of our era, partly due to the Sustainable Development Goals, formulated in 2015 by the United Nations General Assembly and adopted as Agenda 2030. Nowhere is this clearer than in primary production/extraction industries, such as aquaculture and fisheries. Seafood is the world's most widely traded food commodity (Kittinger et al., 2017), and comprises 17% of the world's total global animal protein consumption (I FAO, UNICEF, WFP, and WHO, 2022). In recent years, the pressure for the seafood sector (comprising fisheries and aquaculture industries) to become more sustainable (environmentally, economically and socially) has substantially increased (Portney, 2015; Osmundsen et al., 2020), partly due to increasing public concerns around environmental impact (Olsen and Osmundsen, 2017; Graziano et al., 2018). Yet the dimension we know the least about regarding seafood sustainability is the social; indeed, the problem of how to understand social sustainability has dogged the social science research agenda for decades (Jacobsen and Delaney, 2014). Due to various issues, such as its intangible and qualitative nature, social sustainability is often the vaguest and least explicit dimension (Ballet et al., 2011; Vifell and Soneryd, 2012; Foran et al., 2014; Anderson et al., 2015; Eakin et al., 2017; Béné et al., 2019). Some scholars have tried to unravel the situation (e.g. Vallance et al., 2011), while others have explored a variety of parallel approaches such as corporate social responsibility, the triple bottom line and social licence to operate (e.g. Dahlsrud, 2008; Alibašić, 2018; Alexander and Abernethy, 2019).

Marine and coastal resources provide humans with various economic, nutritional and sociocultural functions (Gilek et al., 2021). Traditionally, fisheries management and governance of coastal and nearshore ecosystems were focused primarily on maintaining biological sustainability within an ecological framework. However, the integration of social sustainability, human dimensions and human rights-based fisheries into international law and conservation policy has increased recognition of the importance of human wellbeing outcomes (Bennett et al., 2021). Advancing democratic ocean governance through recognitional, representational and distributive justice is a key reason for a rise in the study of individual social sustainability loosely covered by the term 'wellbeing' (Gilek et al., 2021). Indeed, wellbeing and quality of life have become what social sustainability seeks to achieve, and the literature in this space has increased significantly in the last decade with most studies attempting to quantify and qualify these concepts (Bravo-Olivas et al., 2015), although it is not clear what change this has yet led to. Wellbeing has traditionally been inferred from economic indicators under the hypothesis that a healthy economic is related to societal wellbeing (ibid.). More recently, however, wellbeing in the literature relates to fairness, equity and justice based on the representation of different groups and individuals in decision-making processes, but also to the consideration of diverging views, beliefs, interests and needs, and how input is weighted (Jacob et al., 2023).

Social sustainability in the seafood sector has tended to focus on wellbeing at the group level, viewing fishers or fish/shellfish/seaweed farmers as a collective, a community, rather than as a collection of individuals (Aguado et al., 2016). This has meant that what Krause and co-authors call a 'people-policy gap' remains (Krause et al., 2015), as social sustainability at the individual level is rarely considered as a goal in and of itself (Cisneros-Montemayor et al., 2021; Jacob et al., 2023). The socio-economic dimensions that make up individual social sustainability in the marine environment include gender, employment and income, nutrition, food security, health, insurance, credit availability, human rights, legal security, privatisation, culture/identity, global trade and inequalities, as well as policies, laws and regulations, the macroeconomic context, political context, customary rules and systems, stakeholders, knowledge and attitudes, ethics, power, markets, capital and ownership (Hishamunda et al., 2009; Alexander et al., 2020; Leposa, 2020; Osmundsen et al., 2020; Gilek et al., 2021). This must be added to the broader community-centred approach to understanding sustainability which includes dimensions such as participation in community and decision-making that affects the community, relationships and trust of others leading to social capital – if we want to understand social sustainability in its broadest sense.

We undertook a 'rapid review' (a review in which design decisions are taken to reduce the time taken to undertake a traditional systematic review) to identify what is currently known about the social sustainability of the seafood sector. There are several limitations to this approach including that the search is less comprehensive, there is non-blinded appraisal and selection, and this may lead to biases in the included articles. We conducted a search using the terms 'social sustainability' AND 'aquaculture' or 'fisheries' OR 'seafood', but we did only search using the English language. We searched in three databases including ScienceDirect, JSTOR and Discovery. Articles were excluded if the term 'social' was mentioned but was not a focus. Using this process, we identified 113 relevant articles. These articles were entered into NVivo and subject to an inductive thematic analysis. Using this approach, we identified seven key thematic areas: livelihoods and human development;

human rights; social, psychological and cultural needs; equitable access to resource and benefit sharing; a voice in public issues; flow-on benefits for local and regional economies and improved infrastructure and access. Despite the limitations of the method, the authors believe these themes to be comprehensive based on expertise of research in this field.

## Livelihoods and human development

Employment is more than just the number of people employed – it can be directly or indirectly related to improvements in quality of life, immigration, demographics, access to/improved health care and consumption of natural resources and includes heritage, lifestyle and healthy living in coastal communities (Aguado et al., 2016; Asche et al., 2018; Gilek et al., 2021). Employment has also been the largest focus for any investigations into social sustainability in the seafood sector.

In fisheries research, it is often noted that jobs are decreasing, due to the inability to bring youth into the sector (Symes and Phillipson, 2009; Tam et al., 2018) and the 'greying of the fleet' (Tam et al., 2018; Donkersloot et al., 2020; Fleming et al., 2020). Alternatively, aquaculture is frequently lauded for its job creation opportunities (e.g. Pierce and Robinson, 2013; Aarstad et al., 2023), however whether this happens at the level industry proposes is debated (Alexander, 2022). Livelihoods are frequently discussed in an artisanal fisheries or local community context against a background of change and increasing vulnerability and the need for communities to diversify and local people to secure jobs (Gilek et al., 2021). In the seafood sector, it is also not uncommon to see family and kin supporting small-scale operations in places such as Alaska (Donkersloot et al., 2020), Brazil (Glaser and Diele, 2004), Cambodia (Larson et al., 2022) and India (Adiga et al., 2016; Apine et al., 2019).

The question of gender, age and race have been explored in the social sustainability literature – particularly in relation to employment (e.g. in social sustainability indicators used by Valenti et al., 2018). Nearly half of the workforce in fisheries is estimated to be female, playing significant but often 'invisible' roles as they may be unrecognised, unpaid and underpaid (Freitas et al., 2007; Zhao et al., 2013). Research has shown that the embeddedness of women in a community affected their wellbeing but did not apply in the same way to fishers from the same community. This is because fishers gain their livelihoods offshore while women typically undertake employment in port such as processing, marketing or accounting alongside others from the community (Nicheva et al., 2022). In aquaculture, women may be restricted to certain roles and face other obstacles such as ill-fitting protective clothing, restrictive maternity leave policies and a lack of investment in support such as childcare in rural areas (Kelling and Lawan, 2023). Employment in coastal sectors suffers from a lack of gender-disaggregated data (ibid.) but from the data that is available, aquaculture and related marine food producing sectors tend to employ a workforce with low education, which is often seen as a key factor for mobility on the labour market, and which makes them vulnerable to social change (Nicheva et al., 2022). To make marine food sectors more attractive for women and to meet social, psychological and cultural needs, requires evaluating the attractiveness of the industry in general (ibid.). This is why several articles voice either general calls for sociocultural data inclusion, or suggest types of data to be included (Bravo-Olivas et al., 2015; Van Holt et al., 2016; Grimmel et al., 2019; Gilek et al., 2021; Nicheva et al., 2022). In regards to age, in the

EU-28 (28 countries within the EU that operate as an economic and political block), every third employee in aquaculture is younger than 40 and in fish processing 42% of employees are less than 40 years old but the workforce is less educated than the overall EU-28 working population (Nicheva et al., 2022).

Fair and equal conditions for all in aquaculture/fisheries and supporting industries regardless of gender, nationality or age is constitutionally embedded in the EU (article 21–23, EU charter 2012) (EU 2012), but social sustainability often receives a lower priority in both policy development and research to fill knowledge gaps (Gollan et al., 2019). As a result, inclusive approaches leading to improved social justice, fundamental to achieving SDGs, are missing (Desiderio et al., 2022).

Improved education and skills training are another important feature of human capital and again often used as a social sustainability indicator (e.g. Valenti et al., 2018; Tiwari and Khan, 2019). However, the role of the seafood sector in this is unclear. The education level of fishers is often below the general population average (Adiga et al., 2016; Apine et al., 2019) and in small-scale fisheries company-led education programmes do not exist although they may do in large-scale fisheries (Van Holt et al., 2016). In many instances, it is noted that training and other non-formal education is often received by those in the seafood sector and that this does increase their skill and consequently the sustainability of the system (Bailey and Eggereide, 2020; Pereira et al., 2021).

## Human rights

For decades, sustainability in seafood supply chains concerned improving traceability from an environmental and food safety perspective, but the 'social side' of traceability was overlooked. Global estimates are that at least 40 million people work under coercive or forced labour conditions across industries such as textile, agriculture, construction and fisheries (Tickler et al., 2018). Human rights abuses such as slavery, forced labour and human trafficking as well as unsanitary conditions, low wages and assault are widespread in fisheries around the world (David et al., 2019; Sparks et al., 2021). Over the past few years, media attention on human rights violations in the seafood industry have grown (Urbina-Cardona et al., 2023), adding to existing persistent ecological pressure from overfishing, illegal, unregulated and unreported (IUU) fishing and climate change (Vandergeest and Marschke, 2020; Wilhelm et al., 2020; FAO, 2022).

Human rights and labour abuses are not restricted to IUU fishing (EJF, 2010; Selig et al., 2022) and plenty of evidence exists of exploitation, lack of safety at sea, overwork, non-payment of wages, bonded labour, unfair recruitment, not fit-for-purpose visa systems, child labour, gender violence and physical abuse (Mackay et al., 2020; Sparks et al., 2021; Willis et al., 2023). Instead, modern slavery is facilitated by the structures within which fishing takes place (Tickler et al., 2018). Human rights abuses are just one of multiple injustices experienced by seafood workers that also includes the undermining or denial of civil, political, economic, social and cultural rights (Bennett et al., 2019; Teh et al., 2019). This has led to institutionalised inequality, collectively driving social instability, poverty and resource decline (Kittinger et al., 2017).

The seafood industry, especially consumer-facing actors such as retailers, use voluntary, non-governmental, market-based governance tools that include ethical standards, 'responsible sourcing' commitments and procurement policies, certification and labelling systems, codes of conduct and guiding frameworks, plus auditing strategies (Sparks et al., 2021), to interpret environmental, human rights and labour laws through consumer facing (B2C) or business-to-business labels. They do this to demonstrate ethical leadership, address risks, protect brand reputation, improve business performance and meet regulatory pressures (Kittinger et al., 2017). Significant investment by private sector actors has set and enforced norms and standards for a multitude of discrete sustainability goals that allows products to be defined as 'sustainable' when owning just one of these characteristics. A truly sustainable product must account for ecological and human wellbeing.

## Social, psychological and cultural needs

In the seafood sector, a variety of aspects that bond a community together and promote social mobility are clear. For example, fishing is often linked to the history and tradition of the places in which it occurs (Reed et al., 2013; Urquhart and Acott, 2013; Ignatius et al., 2019). On the other hand, aquaculture has been perceived as negatively affecting coastal culture and tradition, by causing an employment switch from traditional to new industrial seafood production (Barrett et al., 2002) or by disrupting where indigenous fishing activities can be undertaken (Bailey and Eggereide, 2020). Identity is also strongly influenced by the seafood sector in coastal locations. Fishermen individually often identify strongly with their livelihood, but this is seen at a community level also. As an example, the development of the oyster industry in the Eyre Peninsula in South Australia has been found to strengthen community identity by providing visibility to 'outsiders' and acknowledging community worth (Pierce and Robinson, 2013). The seafood sector has been found to contribute to community spirit and pride (Pierce and Robinson, 2013; Reed et al., 2013), and to connection to place (Jacobsen and Delaney, 2014; Ignatius et al., 2019; Lin and Bestor, 2020). Other aspects of cultural capital which contribute to social sustainability include food provision (Crona et al., 2015; Hornborg et al., 2019; Pereira et al., 2021), amenity value (Lin and Bestor, 2020; Alexander, 2022), local and traditional knowledge (Franco-Meléndez et al., 2021) and spiritualism (Ignatius et al., 2019; Wallner-Hahn et al., 2022).

Whether they are fishers, fish/shellfish farm workers or processors, people in the seafood sector carry out their lives in communities and are members of wider social groups. However, this is an aspect of community capital where much less is known about the influence of the seafood sector. Scholars have noted that choices made around livelihood are rooted in social relationships and community (Donkersloot et al., 2020). In particular, the building of relationships has been identified as a key component of developing a social licence to operate for the seafood sector (Fleming et al., 2020; Billing et al., 2021; Alexander, 2022). A key focus of this area has been around social peace and conflict – often caused by marine stakeholders/users (Glaser and Diele, 2004; Papageorgiou et al., 2021; von Thenen et al., 2021). Issues around intergenerational equity have been recognised but not explored (Halpern et al., 2013; Van Holt et al., 2016; Lisa Clodoveo et al., 2022).

## Equitable access to resources and benefit sharing

Social sustainability adds an emphasis on relational and collective processes to existing fisheries management and governance mechanisms such as equitable access to resources, sharing of benefits and adherence to human rights and labour laws, many of which are currently absent from high profile documents on the Blue Economy

(Armitage et al., 2012; Cisneros-Montemayor et al., 2021). For example, most seafood certification programmes are established by non-public organisations and remain focused on environmental sustainability, rather than social sustainability, equity or fairness (Cisneros-Montemayor et al., 2021).

Socially legitimising these aspects by giving due consideration to local conditions and culture are key to achieving coastal development objectives (Krause et al., 2015; Cisneros-Montemayor et al., 2021; Barreto et al., 2020; Bennett et al., 2021). Without this, blind spots to human behaviour in policies and laws, institutional arrangements and enforcement and compliance are created. This can reinforce command and control approaches, giving less attention to the equitable distribution of benefits. Even less attention may be given to aspects of social equity surrounding the development of marine sectors (Cisneros-Montemayor et al., 2021), leading to more inequalities and vulnerabilities (Barreto et al., 2020).

### A voice in public issues

A sustainable blue economy will only be realised if human wellbeing and justice are placed at its core (Gollan and Barclay, 2020; Bennett et al., 2021; Campbell et al., 2021; Issifu et al., 2023). Nevertheless, ocean policies have been described as equity-blind (Lubchenco and Haugan, 2023), where wealth and power are largely concentrated in certain states as a result of neo-colonial structures or with large corporations. These powerful actors dominate decision-making processes (Bennett et al., 2019, 2022; Hicks et al., 2022), excluding specific worldviews and alternative development pathways (Blythe et al., 2021). When marginalised groups lack a voice in decision-making processes, their needs, perspectives, and rights may be overlooked or disregarded. This can perpetuate inequality, exploitation and unfair practices (Wilhelm et al., 2020; Decker Sparks et al., 2022). This is particularly the case for small-scale fishers, who make up 95% of the world's 4.1 million fishers, and who are often excluded from key decision-making processes, despite contributing to the food security of around 4 billion consumers globally. Bringing human wellbeing and social equity into current ocean governance is the only way to achieve true social sustainability in the future (Cisneros-Montemayor et al., 2021; Blythe et al. 2021).

Across the seafood sector, concerns have been raised regarding the failures of existing regulatory regimes to incorporate local voice (Ignatius et al., 2019Billing et al., 2021; Doerr, 2021; Franco-Meléndez et al., 2021; Alexander, 2022) although it is often unclear how communities have engaged in the process to campaign for change. Equity and justice in decision-making has been a key focus of research, with many scholars proposing that adequate access to the information and tools needed to effectively participate in and influence decision-making is key (Halpern, 2003; Jacobsen and Delaney, 2014; Hadjimichael, 2018). In some cases, those that are affected most by decisions relating to the seafood sector are not involved, for example, crab collectors in a mangrove crab fishery in Brazil were found not to be included in the planning and implementation of fishery management (Glaser and Diele, 2004). The need to include local knowledge and local perspectives to increase social sustainability has also been noted (Jacobsen and Delaney, 2014) and indeed was found to improve social sustainability for Territorial Use Rights for Fisheries in Chile (Franco-Meléndez et al., 2021).

More often than not, too little attention has been given to the ways individuals will gain, lose or be excluded from coastal development. Most social sustainability issues are not considered in isolation which arguably reflects the multifaceted nature of social sustainability as well as the mix of analytical and normative approaches in the literature (Gilek et al., 2021). However, they all point to the power and role of human agency (Armitage et al., 2012), covering the areas of wellbeing; livelihoods and human development (material assets and basic needs); and social, psychological and cultural needs (Gilek et al., 2021). Power asymmetry through exclusionary processes tend to legitimise predetermined outcomes, which can decrease recognition and representation (ibid.). When participation in management increases, the wellbeing of society is also improved (Datta et al., 2012). Recognition of diverse social and cultural values and different forms of knowledge is key, and emphases the close connection between different aspects of social sustainability (Gilek et al., 2021). However, even when people affected are clearly identified, a policy for systematically including them in often lacking (Krause et al., 2015).

### Flow-on benefits for local and regional economies

Often of interest regarding social sustainability of the seafood sector is the input that it has into local, regional, and national economies – all of which contribute tax income to local, regional, and national governments. Benefits to the economy are common in industry and government discourses, although resistance movement discourses suggest that this effect tends to be over-exaggerated (Crona et al., 2015; Alexander, 2022). It seems that the research results are mixed. For example, a study of seaweed culture has shown that over half of the investment and operating expenditure is spent in local markets (Pereira et al., 2021). It has also been argued that in Taiwan the Bluefin Tuna Cultural Festival – directly linked to the fishery – has increased economic prosperity (Lin and Bestor, 2020). However, in Canada, it has been shown that community and regional economic benefits are not automatically derived from simple quota allocations, but instead depend on a variety of factors (Foley et al., 2018). Furthermore, while the seafood sector has been viewed as a vehicle to alleviate poverty (Crona et al., 2015; Bush et al., 2019), small-scale fisheries are often one of the poorest and most vulnerable groups worldwide (Apine et al., 2019) and there is little evidence of how much of an effect the industry has. Some studies, however, have suggested a change to poverty level by those working in the sector (e.g. Glaser and Diele, 2004; Bush et al., 2019).

### Improved infrastructure and access

Human-constructed infrastructure that supports society is the least explored area of seafood social sustainability. The need for basic local services such as schools, medical facilities, public transport and affordable housing is frequently noted (e.g. Symes and Phillipson, 2009; Apine et al., 2019; Larson et al., 2021). However, there is little evidence to suggest that the seafood sector supports such things – at least in the peer-reviewed literature – although a study of the benefits of seaweed farming to wellbeing in Indonesia suggested that there had been improvements to housing and health (Larson et al., 2021). In the grey literature, an assessment of the benefits of aquaculture to Scotland (Alexander et al., 2014), for example, found that the aquaculture industry did directly support housing and internet infrastructure, amongst other things. However, unless the infrastructure is paid for directly by the sector, it is difficult to assess the role that the sector has played in any changes to infrastructure.

## Integrating social sustainability into existing industry and governance processes

Existing industry and governance processes could be strengthened and enhanced in several ways (Figure 1). This may include incorporating social sustainability metrics and indicators that capture relational and subjective aspects into planning and monitoring frameworks, particularly co-created measures that relate to community wellbeing, social equity and inclusivity. Current social impact assessments can be constrained by quantitative measures, integrating qualitative aspects such as community empowerment and capacity building efforts would enable full participation of all relevant stakeholders. Indeed, ongoing dialogue between industry stakeholders, government bodies and communities of interest/ place should be established to foster meaningful engagement and ensure social considerations are embedded in decision-making at every level. Mechanisms established for community participation and the inclusion of local voices could include community advisory panels, promoting community-based management approaches and fostering partnerships between industry and local stakeholders. Community engagement protocols should be co-developed. Responsible business practices, procurement guidelines and standards could include initiatives that enhance the overall quality of life at value chain 'touch points', with standards and certifications that explicitly include social criteria that address the breadth of social sustainability. Transparent reporting on social sustainability practices will lead to change, including mandatory requirements for companies to disclose their social impact, community engagement initiatives and adherence to social sustainability criteria. Using digital platforms can help with that reporting while learning platforms can facilitate knowledge exchange and collaboration to enhance transparency and accountability. Finally, regular evaluation of initiatives to assess effectiveness and best practice will enable adaptation to evolving social and environmental contexts.

## Conclusion

The incorporation of social equity concerns into sector management is required to ensure human security amidst ongoing global challenges (Gollan et al., 2019). Seafood policy that also considers economic and social sustainability in addition to environmental will increase the likelihood of greater social acceptance of management policies and priorities (ibid.). It is clear, however, that several knowledge gaps remain. While there has been a clear focus on developing social sustainability indicators, this has focused on those aspects which are easily measurable such as numbers of jobs and demographics. Approaches to identify and determine the effects of the more relational and subjective aspects of social sustainability (see e.g. Fudge et al., 2023) remain unclear. Some thematic areas of social sustainability also remain underdeveloped including fairness and equity in resource sharing and benefits, flow-on benefits for local communities and the role that the seafood sector plays in infrastructure and access to services. Moreover, the concept of the right to food as a human right was not explicitly mentioned in the literature (nor is it often considered within the legislation – e.g. UK Human Rights Act 1998), but this could be an area worth further consideration. A truly sustainable seafood product must account for tenable ecological and human wellbeing. If not, institutionalised inequality and compromised food, resource and livelihood security are just some of the outcomes – a state in which, it could be argued, we find ourselves today. There is an acute need to address the knowledge gaps identified above and incorporate what we already know about social sustainability into existing industry and governance processes. Without doing this, it will be impossible to meet the Sustainable Development Goals, in particular targets relating to zero hunger and sustainable food systems (T2.4); equal and safe employment (T8.5, T8.8); reduced inequalities (T10.2, T10.4); sustainable management of natural resources (T12.2, T12.6, T12.8) and transparent and participatory decision-making (T16.6, T16.7).

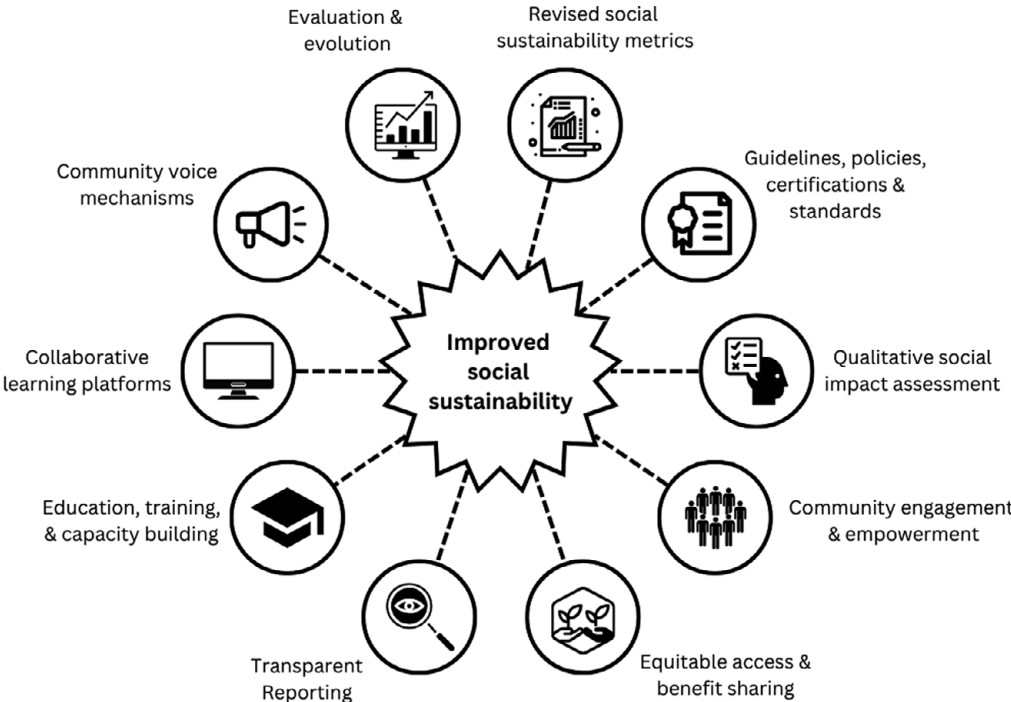

**Figure 1.** Mechanisms by which to incorporate social sustainability into governance and industry processes.

To not do this may mean moving the dial even closer towards environmental and societal collapse.

**Open peer review.** To view the open peer review materials for this article, please visit http://doi.org/10.1017/cft.2023.31.

**Author contribution.** Conceptualization: K.A.A.; Data curation: K.A.A. and I.K.; Formal analysis: K.A.A. and I.K.; Writing – original draft: K.A.A. and I.K.; Writing – review & editing: K.A.A. and I.K.

**Competing interest.** The authors declare none.

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
