## [Reviewer Report]

Social sustainability in seafood systems

This is an interesting article that has great potential to contribute to this research area and to help guide action in the seafood sector. My main comment – and source of request for major revision – is that it is unclear from the start what type of review this is. Can the words ‘literature review’ be added to the title? Also, in the last paragraph of the introduction there needs to be some further explanation of what type of review this is (presumably it was a scoping literature review) and how it was conducted. Following that should be a guide to the rest of the review including how the 7 key thematic areas mentioned in the abstract were conceived – were these major thematic areas identified in the review? Or where they developed by the authors and matched to the literature?

Detailed comments

Impact statement

– ‘regarding the term’s many meanings’ – unclear exactly what ‘the term’ is? Do you mean social sustainability? Or sustainable development?

Also the lines on benefit of the work – agree this research will be useful to inform scholars and potentially those working in the sector, although I wonder if those working in the sector may already be aware of the challenges but struggle with implementing change? A section on overcoming this challenge, particularly examples from industry, would enhance the review.

Abstract

- 7 key thematic areas mentioned here and nowhere else

Introduction

Seafood sector (comprising fisheries and mariculture industries) – unclear if freshwater included or if excluded, why?

Unravel the chaos – choice of words potentially a bit over the top! Or else needs some more detail to elaborate on the chaos.

‘The integration of social sustainability…has increased recognition of human wellbeing outcomes’ – this seems at odds with you argument. Where have the integration of these concepts increased? And where has recognition of human wellbeing been increased?

‘literature in this space has increased significantly..’ – agree but can you comment on if/how this has resulted in changes in the sector itself? Or influenced industry?

‘The socio-economic dimensions that make up individual social sustainability…’ – this is a very long list of dimensions that only seem to come from one reference, but on checking they do not appear in that reference which focusses mainly on freshwater aquaculture. A more rounded list of dimensions should be sourced from a wider range of sources.

Last paragraph of introduction – as mentioned earlier, more detail is required here on the review method and the structure of results.

Livelihoods and human development

‘whether this happens at the expected level is debated..’ – what is the expected level? Also author name for reference not displayed correctly here and throughout document

EU-28 – will all readers automatically understand what this is?

Human rights

- No mention of the right to food?

Equitable access

- The last sentence is too long and should be split

A voice in public issues

‘A sustainable blue economy will only deliver’ – deliver what?

Conclusion

‘incorporate what we already know about social sustainability into existing industry and governance process’ – agree. As suggested above, is there scope to build on this here, or in the main part of the document, and suggest how? What do the successful examples you reviewed have in common? May be beyond scope but any advance on articulating a need through ideas on how to address it would enhance the review greatly (and help avoid environmental and societal collapse…).

---

## [Reviewer Report]

This manuscript reviews literature regarding social sustainability of the seafood sector. The review is very well written and the emerging themes are clearly presented and described. The material presented appears comprehensive although without seeing the details of the methods conducted, it is not clear how the literature was selected and analysed. The lack of a methods section appears to be the accepted format for this journal, therefore it seems no revisions are required in this respect. However, it would be useful to include come insightful comments about the limitations of the literature reviewed, if any (geographical distribution, missing themes, etc) to better situate the conclusions. I also recommend including key comments and suggestions about how the themes presented and the knowledge gaps identified relate to specific SDGs and an interpretation of how filling the knowledge gaps in concert with the application of current knowlege about social sustainablity could help avoid the environmental and social collapse mentioned in the closing sentence.

---

## [Editor Report]

This is a very well written and constructed review. Key questions raised by Reviewer 1 need to be given due attention included the connection to the right to food as a human right. Reviewer 2’s challenge for an explanation of how the themes identified relate directly to SDG delivery will provide an important rounding of the conclusion and provide space for the authors to articulate what a successful integration of social sustainability may look like in different cultural, geographical and economic contexts.